# Impact of productive safety net program on food security of beneficiary households in western Ethiopia: A matching estimator approach

Aregash Getachew Hailu[1]*, Zerihun Yohannes Amare[2]

**1** United Nations World Food Program, Gambella Sub Office, Gambella, Ethiopia, **2** Institute of Disaster Risk Management and Food Security Studies (IDRMFSS), Bahir Dar University, Bahir Dar, Ethiopia

* haregget2006@gmail.com

**Data Availability Statement:** All relevant data are within the manuscript and its Supporting information files.

## Abstract

From various Ethiopian government food security strategies, the Productive Safety Net Program (PSNP) is one of the strategies to improve households' consumption. As a result, the government needs to know the outcome of the program intervention for further decisions in similar poverty reduction strategies. This study examined the impact of Productive Safety Net Program intervention on food security of rural households in rural western Ethiopia. A cross-sectional survey data were gathered from a total of randomly selected 188 beneficiary and non-beneficiary sample households. Key informants interviews and focus group discussions were employed to triangulate household survey results. A Chi-square test was employed to compare the households' food security status. The propensity score matching method was used to evaluate the impact of a Productive safety net on beneficiary households' food security status. In this study, the Productive Safety net program has significantly increased households' calorie intake. The beneficiaries' households were more food secure than non-beneficiary households by 68% and 54% respectively. The mean energy available for the beneficiary and non-beneficiary households is 2488.500 and 2153.394 kcal, respectively. Estimates of the average treatment effect of the treated indicated that a Productive safety net program can impact households food security by 2519.29348 kilocalories, higher in the kernel matching method. This is an encouraging indicator for Programme implementers and funding agents. Therefore, channeling further efforts on this indicator is important for a more pronounced impact of the Programme. Given a certain level of variations of different program impact studies, this study recommends further research with greater scope and at different locations on the impact of PSNP and related Food Security Programs on households' food security status.

## Introduction

Food insecurity is a persistent concern for many countries in the world [1]. The sustainable development goals (SDGs) cover all social, economic, and environmental wings of

**Funding:** The author(s) received no specific funding for this work.

**Competing interests:** The authors have declared that no competing interests exist.

sustainability. Specifically, SDG-2 aims to End hunger, realize food security and enhanced nutrition and promote sustainable agriculture [2].

Ethiopia is among the severely food-insecure, drought and famine-affected nations in Sub-Saharan Africa. The majority of Ethiopia's nations are affected by persistent and cyclical food insecurity [3]. Smallholder agriculture has long served as the dominant economic activity for people in sub-Saharan Africa [4], and this is also true in Ethiopia.

The rural smallholder farmers are the most affected [5] where a large number of households cannot produce or buy enough food to cover their annual food needs even under normal weather and market conditions [6]. Because of this problem, the Ethiopian Government in partnership with global donors introduced the Productive Safety Net Programme (PSNP) in 2005. The aim is to link direct relief with development and focus on reducing vulnerability and create enabling conditions for the rural poor households [7]. There are different studies on the impact of PSNP impact of PSNP on child's time spent on schooling and work like [8], saving of assets [9, 10]. However, none of these studies have studied the impact of PSNP on households consumption. Thus, the objective of this study is to assess the impact of a productive safety net program on the food security of beneficiary households in western Ethiopia. Specifically, the study will answer two research questions: how are the existing food security status of beneficiary and non-beneficiary households and how is the impact of the PSNP intervention on the beneficiary households in western Ethiopia.

## Literature review

Globally, 821 million people are undernourished. Of these 257 million are in Africa, of whom 237 million are in sub-Saharan Africa [11]. Ethiopia has suffered a long history of successive droughts and famines in its modern history. The years; 1952, 1959, 1965, 1972, 1973, 1978, 1984, 1991, 1994, 1999 and 2002 were dry years [12]. Almost 8 million people are expected for food aid every year and millions more suffer from seasonal food shortages and malnutrition. To overcome the problem of food insecurity in the country, the Ethiopian Government in collaboration with a consortium of donors introduced the Productive Safety Net Programme (PSNP) in 2005. The objective is to link relief with development and focus on reducing vulnerability and [13] create enabling conditions for the rural poor.

There are similar studies in the PSNP at Kebribeyah *District* of Somali Regional State of Ethiopia [14], and in Bale Zone [15]. However, these studies used only PSNP participants before the intervention and after the intervention of the program without evaluating the non-beneficiaries food security status leading to the counterfactual problem. However, the same respondents recall is often incorrect since it is hard to remember all past incidents correctly, resulting in over or under-reporting of past incidents that lead to recall bias [16]. Besides, these previous studies did not control the other factors that could affect the food security status of the households. This is because it relied on simple descriptive statistics alone. The majority of previous studies on the impact of PSNP was also conducted at the country level to assess its general impact [17–19]. Whereas, at the local level, there is inadequate empirical evidence on the program impact on the expected outcomes, particularly on the Productive safety net program beneficiary households' food security. The main research question for this research is how is the impact of productive safety net program on the food security of beneficiary households.

There is difficulty in finding reliable data on food security which is having the problem for policymakers and researchers [20]. Therefore, the results of this research will contribute to a growing project impact evaluation literature in at least two ways. The first input of the article

identifies the causal effect of a comprehensive food security program on household's consumptions in similar income settings in different parts of Ethiopia.

The second contribution of the results of the research is its use of major determinants variables of food security status of the study households which could be documented as evidence for the diverse intervention of the poverty reduction programs. The primary empirical data sources for this study were a cross-sectional survey of 2019 households from Western Ethiopia, Oromia Regional State, *Sire* District.

## The food security profile of Ethiopia

For several years in the past, Ethiopia has been known as a country that is highly dependent on emergency food aid to its widespread and persistent food insecurity. This humanitarian aid was estimated at US$265 million per year on average between 1997 and 2002 and saved many lives. However, the program end-line impact evaluations have shown that it was unpredictable for both planners and households, and often arrived too little for the target. The setbacks and irregularities of the program meant that the humanitarian aid could not be used effectively and did little or no to protect household livelihoods, maintain environmental degradation, generate communal resources, and household assets [21].

Before the inception of the PSNP, aid schemes such as emergency food aid, Food-for-Work, and Employment Generation Scheme were used as social protection programs in Ethiopia [22]. However, after the severe droughts of 2002/03, there was extreme hunger in Ethiopia and this leads the government to decide to supplement the existing poverty reduction response systems, and the Employment Generation Scheme, with more certainty and sustainable solution to end poverty and reduce the vulnerability of households to food insecurity. To this end, the Food Security improvement Program with different components was endorsed in 2005 under the Ethiopian Ministry of Agriculture and Rural Development [22]. In particular, in collaboration with development partners, the government of Ethiopia initiated a component of food security program with the name Productive Safety Net Program [8] in the same year as a social protection program that makes people's livelihoods more secure [23, 24].

## Program description

The Ethiopian productive safety net program was aimed to cover around two hundred sixty-three districts in the four regions, namely Amhara, Tigray, Oromia and Southern Nations Nationalities and People's, that had been significant recipients of food aid between 2002 and 2004 and operate as a safety net by providing transfers to 4.5 million beneficiaries via either pubic work or direct support [25]. A recent report shows that around 7.8 million eligible households in the country are enrolled in the program [26]. The PSNP provides a minimum of five days of payment per month for six months when there is a low agricultural production season for at least the next five years. A member of the targeted household for the public work employment gets 50 birr9 (US$2.80) or 15kg of grain per month [27]. The type of transfer can be in kind (food), in cash or can be a combination of both; it depends on the transfer that the donors have made. Nonetheless, the transfers are set at a level to smooth out food insecure households consumption to close the food gap over the annual food shortage months. However, due to the high inflation rate, adjustments to the wage rates were made over the period of the program and participants received Birr 8 in 2008 and Birr 10 per day in 2010 [28].

Agriculture is the main engine of Ethiopian Economy which includes crop production and livestock [29]. Since the main aim of PSNP is to safeguard a minimum level of food consumption and enhance livestock accumulation of the needy households, the beneficiaries are expected to graduate from the program once they have achieved better livelihoods and become

food secure. The support from another component of food secure program, namely household asset building program that provides microcredit and agricultural extension services to diversify income sources and increase productive assets of the participants will also continue after graduation [25, 28].

Program impact evaluation is the process of examining the achievement level of a given poverty reduction program and its desired changes to the participants. It is aimed at sorting out the net impact of a program intervention on the participants that can be attributable solely to that specific intervention. It is the act of assessing outcomes in the short, medium or long term change due to an intervention [30]. Program impact evaluation is a matter of studying whether the changes in the well-being of households are certainly due to the program intervention and not to other factors [31]. Sire *District* is among the identified *Districts* for productive safety net program intervention. The program was launched in the *District* in 2006 and no known assessment of the intervention has been made. This study was initiated to assess the impact of the Productive Safety net program for further policy recommendation in Ethiopia and countries having related poverty reduction strategies.

## Methodology

### Description of the study area

Ethiopia is the 27th largest country in the world with a total size of 1,126,829 square kilometers. Its neighboring countries in East Africa include Sudan and South Sudan to the west, Djibouti and Eritrea to the north, Djibouti and Somalia to the east, and Kenya to the south [32].

This study was conducted in Sire *District* in the Oromia region, which is among the ten regional states in Ethiopia. The study area was delineated into lowland, mid-land and high land. The study *District* has two rainy seasons: a long rainy season, locally called *kiremt* (June to August); and a short rainy season, locally called *belg* from March to May. *Kiremt* is the major growing season in the study area. Mixed farming dominates the livelihoods of the community in the study area [33] which is similar to the major parts of the country [34, 35].

The greater majority of the households living in the *District* is engaged in subsistence mixed crop production and livestock rearing. The main food sources for households in this livelihood area are rain-fed agriculture products such as wheat, barley, bean and maize, livestock products like milk, butter, meat [33]. Eighty-Seven percent of the population reside in rural *Kebeles* (The lowest administrative unit next to district) where this study was conducted. Sire *District* has 25 rural *kebeles*. The *District* has a total population of 102,447, out of which 51,019 were males and 51,428 were females [36].

### Methods and sources of data collection

This empirical analysis was implemented using primary data which were collected through a scheduled survey of 188households. A three-stage sampling technique was used to select the final sampled households. Firstly, *Sire District* was purposively chosen based on its wide coverage of the PSNP and vulnerability of communities to food insecurity. In the second stage, among a total of 13 PSNP beneficiary *kebeles*, *Bele*, and *Allu Kebeles* were randomly selected. The households in these *Kebeles* were classified into 2 strata (beneficiary and non-beneficiary). Finally, by using a proportionate sampling method, 94 beneficiaries and 94 non-beneficiary households were selected using simple random sampling procedure. The list of household heads for both beneficiary and the non-beneficiary groups were obtained from the lists available in each of the selected *kebeles* as as a sampling frame. The study used samples of beneficiary households as the treatment group and non-beneficiary households as the control group.

Data was collected through a structured questionnaire. The household questionnaire was primed in the English language and then interpreted into the local language. A pre-test was done by taking 5% of the total sample size from both PSNP beneficiaries and non-beneficiary households. The pre-test participants were out of the target *Kebeles* and not involved in the actual survey. Necessary amendments were made based on the comments obtained from the pre-test schedule survey. Four enumerators who are experienced in the socioeconomic survey were employed and introduced to the aim and purpose of the study. The actual schedule survey was carried in November and December 2019. Secondary data was also collected from relevant office records as well as published and unpublished sources. These include the study *District* office reports, *District* performance reports, and central statistics agency publications.

## Designing the outcome variables: Household calorie intake

Food security has a multidimensional character [37]. This implies that the identification of relevant food security indicators is difficult. Outcome indicators are proxies for food consumption measured either directly as food expenditure and caloric intake or indirectly through anthropometric indicators [38]. Whereas, process indicators reflect the status of food security and the level of vulnerability to food insecurity [39]. The calorie intake indicator is the principal variable used to define food poverty by the Ethiopian government [40]. In this study, food calorie intake was employed as an outcome variable.

A household schedule survey was used to construct the food calorie intake variable at the household level. Respondents were asked to recall food types consumed from home production, purchases and or gift loans/wages in kind and the amount the last fourteen days preceding the household survey. The physical food amount consumed by a household was converted into calories. This was also adjusted for household age and sex and followed four steps. Firstly, the local measurement units were converted into a common unit of measurement for each food item consumed by a household. Secondly, the food items consumed by households were transformed into calories using the Ethiopian food composition table designed by National Health and Nutrition Research Institute [41]. Thirdly, food calories consumed by households were summed up and converted into daily amounts. Lastly, the combined food calories were adjusted in an adult equivalent unit per household.

The amount of energy in kilocalories available for the households was recorded then the results weighed against the minimum requirement per adult equivalent per day 2100kcal. Those households who consumed below minimum requirement were classified as food insecure while those beyond and equal to the threshold were categorized as food secure. The households calorie intake was compared with the national average daily caloric intake requirements for a moderately active adult ($\leq$ 2100 kcal) to know the food security status of the productive safety net program participant households in the study area. Based on this information, those households who met the above-estimated caloric requirement were classified as food secure and otherwise as non-food secure.

## Data analysis

The data entry was done by using excel and exported into STATA (version15) and then the analysis of data was undertaken through descriptive statistics and econometric models. Descriptive statistical techniques such as mean, percentage, and standard deviation were applied to describe the socio-economic characteristics of respondents. In addition to this, descriptive tools such as charts and tables were illustrated to present data.

**Specification of Propensity Score Matching (PSM) model.** The pros and cons of the different non-experimental methods for impact evaluation were discussed by Blundell and Costa

**Table 1. Variable description and measurement.**

| Variable | Type and description | Measurement |
|---|---|---|
| | | 1 if yes,0 otherwise |
| Dependent variable: TREATMENT Independent variables: | Dummy, participation in PSNP | |
| AGEHH | Continuous, age of household head | Number of years |
| SEXHH | Dummy, Sex of household head | 1if male,0 otherwise |
| FAMSIZE | Continuous, family size | Number of household members |
| LAB FORCE | Continuous, adult labour force of age 15–64 years | Number of the active labour force |
| EDUHH | Continuous, education of household head | Years completed |
| LADSIZE | Continuous, total land size | Hectare |
| NONACT | Dummy, non-farm activities | 1 if yes,, 0 otherwise |
| TOTAL-TLU | Continuous, total livestock | Tropical livestock units |
| OXN | Continuous, Oxen owned | Number of oxen |
| ACCEXT | Dummy, access to extension service | 1 if yes,0 otherwise |
| ACCCRD | Dummy, access to credit | 1 if yes,0 otherwise |
| NUMBMO | Continuous, food gap faced | Number of months per year, |

Source: Household Survey 2019.

Dias [42, 43]. In our study area, there was no baseline data before the program intervention of the PSNP. Therefore propensity score matching was employed which is usually ideal to evaluate the impact of productive safety net program on the beneficiaries when there is no baseline data [44]. When there is baseline data and follow-up survey data, a more robust procedure for impact evaluation would be difference in difference (DIDI) in combination with PSM [42, 43]. Thus, this study applied a propensity score matching (PSM) which could identify comparable treatment and comparison controls using cross-sectional household data based on [45].

*Choosing a matching algorithm.* Despite the existence of plenty of methods to do so, only three of them get attention [46]. Three matching algorithms commonly used, namely nearest neighbour matching, radius matching, and kernel-based matching were implemented to evaluate the impact of PSNP on households food security to match the treated and control observations. To choose the best matching estimator for the analysis, different guiding criteria based on Dehejia and Wahba like the equal means test referred to as the balancing test, low Pseudo $R^2$, and high matched sample size were taken into consideration [46]. Finally, kernel matching was used for this study analysis.

This study tried to maintain essential pre-conditions suggested by Heckman and Ichimura [47]. As a result, data were calculated using the same questionnaires for both treated and untreated groups, treated and comparison controls share similar socioeconomic, demographic and agro-ecological settings. Relevant variables related to treatment and outcome were included in the PSM function(Table 1).

## Results and discussion

Tables 2 and 3 show, PSNP and non-PSNP households had a statistically significant difference in terms of cultivated land size, total livestock unit, household food gap in a month, marital status, credit access. Non-beneficiary households are better off on these factors than beneficiary households. This is plausible since most of the beneficiaries were targeted for PSNP didn't have large farmland and major livestock assets. However, the number of family size, the

**Table 2. Summarized descriptive statistics of sample households.**

| Variables | PSNP households | | Non-PSNP households | | Difference in means | t- value |
|-----------|------|------|------|------|------|--------|
| | Mean | STD | Mean | STD | Mean | |
| Age of HH | 45.106 | 18.2229 | 40.989 | 13.6417 | -4.1170 | -1.754 |
| HH Family size | 4.202 | 1.7387 | 4.106 | 2.0079 | -.0957 | -0.3495 |
| land size (ha) | 0.7223404 | .468948 | 1.031915 | .8872863 | .310 | 2.9118*** |
| Active labor force | 2.617021 | 1.201311 | 2.87234 | 2.31514 | 0.255 | 0.9491 |
| (TLU) | 1.628 | 1.8138 | 2.723 | 3.3098 | 1.0957 | 2.815*** |
| Food gap in month | 4.234 | 1.2904 | 3.351 | 1.4196 | -.8830 | -4.4625*** |

***, and * stand for significance at the 1% and 10% levels, respectively.

Source: Computed based on Household Survey 2019.

household age in year and availability of labour force, there was no statistical difference between the two groups.

## Households' food security status

This study objective is used as a springboard to study the impact of PSNP intervention on beneficiary households. 42% of households were found to be unable to meet the minimum survival requirement or food insecure and 58% of households were found to meet the minimum energy requirements or food security (Table 4). The result also shows that the percentage of food secured beneficiaries households and food secured non-beneficiary households are 68% and48% respectively. Therefore, the greater part of food-insecure households was found to be in non-beneficiary households. A Chi-square test was employed to compare the beneficiary households' food security status.

The Pearson chi-square is found to be p<0.01 which is significant at a 1%level of significance (Table 5). The mean energy available for the beneficiary and non-beneficiary households is 2488.500 and 2153.394 kcal, respectively. So, the t-value reveals that there is a significant mean difference between the beneficiary and non-beneficiary households at a 1% level of

**Table 3. Categorical variables chi-square test.**

| Variable | | Non-PSNP participant | PSNP- participant | Ch2 | P-value |
|----------|--|------|------|------|------|
| | | N (%) | N (%) | | |
| Gender of the household head | Male | 71(75.53) | 60(63.83) | 3.0465 | 0.081 |
| | Female | 23(24.47) | 34(36.17) | | |
| Marital status | Single | 4(4.26) | 10(10.64) | | |
| | Married | 79(84.04) | 56(59.57) | 13.9002 | **0.001*** |
| | Divorced | 11(11.70) | 28(29.79) | | |
| Education | Non-educated | 57(60) | 57(60) | | |
| | Educated | 37(40) | 37(40) | 0.0000 | 1.000 |
| Extension access | Yes | 71(75) | 77(82) | 1.1432 | 0.289 |
| | No | 23(25) | 17(18) | | |
| Credit access | Yes | 22(23) | 40(42) | 7.7972 | **0.005*** |
| | No | 72(77) | 54(58) | | |

* stand for significance at the 1%.

Source: Computed based on Household Survey 2019.

**Table 4. Proportion of sample households' food security status.**

| Households food security status | | | |
|---|---|---|---|
| | Food secure(n = 109) | Food insecure(n = 79) | total(n = 188) |
| Beneficiary | 64 | 30 | 94 |
| | 68% | 32% | 100% |
| Non-beneficiary | 45 | 49 | 94 |
| | 48% | 52% | 100% |
| | $X^2 = 7.005^{***}$ | | |

Where *** indicated significance level at 1%.

Source: Computed based on Household Survey 2019.

significance (Table 5). Thus, the result reveals that there is a significant association between PSNP interventions and the food security status of households.

## Impact of PSNP on food security

The first part of the econometric analysis is the propensity score matching analysis to investigate the causal effect of the Productive safety net program on households food security. The balancing property is tested to confirm that individuals with the same propensity score have the same distribution of observable characteristics (Table 6).

## Common support condition and bias reduction

To estimate the impact of PSNP on household food security predicted propensity score is used to match farmers with similar characteristics to compare the impact of PSNP on households' food security. Only observations in the common support region matched with the other group considered and others should be out of further consideration(Rosenbaum and Rubin, 1983). Once the region of common support is identified, sample households that fall outside this region were dropped and the treatment effect cannot be estimated for these sample households. The region of common support selected was [.13898677, .97527122]. The common support region would then lie between .13898677 and .97527122 are not considered for the matching exercise (Fig 1). Moreover, this study applied a visual analysis of the density distribution of the propensity score in the two groups (treated and non-treated groups) to check overlap and common support before matching samples. The common support assumptions imply that the probability of receiving treatment for each possible value of the vector X is strictly within the unit interval that falls outside the region of the common support area would be dropped.

**Table 5. Energy available per adult equivalent per day for sample households.**

| Energy available per AE in (Kcal) | PSN P beneficiaries | | Non- PSN P | | Mean. Difference | T- value | P-value |
|---|---|---|---|---|---|---|---|
| | Mean | SD | Mean | SD | Mean | | |
| | 2488.500 | 1159.9464 | 2088.543 | 855.5304 | -399.9574 | $-2.690^{***}$ | 0.008 |

Where *** indicated significance level at 1%.

Source: Computed based on Household Survey 2019.

**Table 6. Estimation of propensity score through a probit regression model.**

| Treatment variable | Coef | Std.err | Z | p>z |
|---|---|---|---|---|
| Gender | -3417147 | .2510744 | -1.36 | 0.174 |
| Marital status | -1238461 | .2155667 | -0.57 | 0.566 |
| Age | .0252931 | .0080197 | -3.15 | 0.002 |
| Education | .431874 | .2356755 | 1.83 | 0.067 |
| Family size | .0134191 | .0646188 | 0.21 | 0.835 |
| Labour | -0393968 | .0619658 | -0.64 | 0.525 |
| Land size | -3989577 | .1642282 | -2.43 | 0.015 |
| Oxen number | .0244243 | .1427754 | 0.17 | 0.864 |
| TLU | -0853543 | .046574 | -1.83 | 0.067 |
| Access to credit | .4593664 | .227838 | 2.02 | 0.044 |
| Access extension | .4600552 | .2702364 | 1.70 | 0.089 |
| Off-farm | .5382483 | .22878332 | 2.35 | 0.019 |
| Food gap in month | .237187 | .0778764 | 3.05 | 0.002 |
| Con | -2.186331 | .797949 | -2.74 | 0.006 |
| Probit regression<br>LR chi2(13) = 56.11<br> Probit > Chi2 = 0.0000<br> Pseudo R2 = 0.2153<br>Log Likelihood = -102.25886 | | Number of observation = 188 | | |

Source: Computed based on Household Survey 2019.

## Choosing a matching algorithm

Despite the existence of plenty of methods, this study employed the three PSM estimators based on authors such as Deheija and Wahba, Caliendo and Sabir [46, 48]. Matching estimators like nearest neighbour matching, radius matching, and kernel-based matching algorisms were used to evaluate the impact of PSNP based on Leuven and Sianesi [49] with STATA15.

To choose the best matching estimator for the analysis different guiding criteria, such as equal means test referred to as the balancing test, low Pseudo $R^2$ and matched sample size were taken into consideration. Thus, a matching estimator which balances all the explanatory variables that result in insignificant mean differences between the two groups, bearing low pseudo $R^2$ value and results in a large matched sample size was taken as the best estimator. As it is illustrated in Table 7 kernel with all bandwidth estimators have resulted in the lowest pseudo value, well-balanced covariates and largest sample size by discarding only two households (2 treated and 0 control households) from the sample. Hence, only the results obtained from this estimator were presented and discussed.

## Balancing test

The balancing test of covariates tests the significance of the mean difference between the matched and unmatched samples in terms of all the covariates used for the matching purpose. The unmatched samples of the beneficiary and non-beneficiary households were significantly different in terms of certain characteristics. However, one looks to see that any differences in the covariate means between the two groups in the matched sample have been eliminated, which would increase the likelihood of unbiased treatment effects.

The calculated test result measures the balancing of the distribution of p-value, for each variable used in the regression; it calculates the p-value for equality of means in participant and

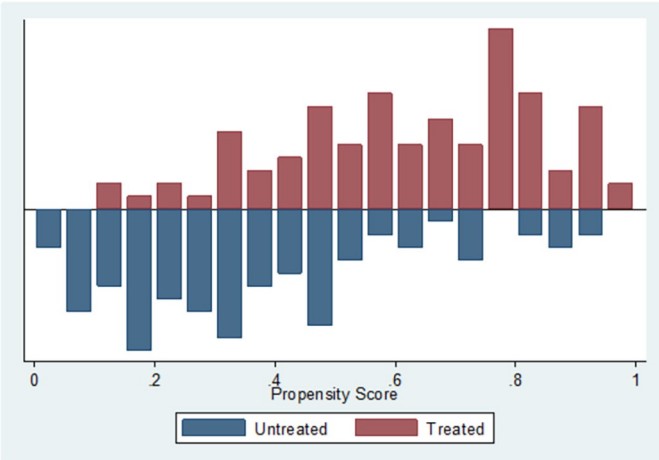

**Fig 1. Common support region graph.** Source: Computed based on Household Survey 2019.

non-participant groups, both before and after matching. The p-value is based on a regression of the variable on the participant indicator. Before matching this is an un-weighted regression on the whole sample while after matching the regression is weighted using the matching weight and is based on the support sample. According to the p-value of individual tests, similarities in the mean values between treatment and control groups in this matching estimator, relatively all of the variables have lower p-value (insignificant). This shows kernel bandwidth matching is preferred as the best estimator of the average treatment effect. Consequently, only the outcome of this estimator is used to meet the study objectives of estimating the impacts of PSNP on the food security of households(Table 8).

Table 9 indicates that the value of Pseudo− $R^2$ was very low. This low Pseudo− $R^2$ value and the insignificant likelihood ratio test indicate that the PSNP participant and nonparticipant household had the same distribution in the covariates after matching. These results indicate that the matching procedure can balance the characteristics in the treated and the matched

**Table 7. Performance of matching estimators using three criteria.**

| Matching Estimator | Performance criteria | | |
|---|---|---|---|
| | Balancing test | Pseudo $R^2$ | Matched Sample size |
| **NN matching** | | | |
| NN matching without replacement(1neigboure) | 8 | 0.2177 | 138 |
| NN matching with replacement(1neigboure) | 8 | 0.2177 | 185 |
| **Radius caliper** | | | |
| With 0.01 band width | 8 | 0.2177 | 149 |
| With 0.25 band width | 8 | 0.2177 | 186 |
| With 0.5 band width | 8 | 0.2177 | 186 |
| **Kernel matching** | | | |
| With 0.01 band width | 8 | **0.2177** | **186**[*] |
| With 0.25 band width | 8 | **0.2177** | **186**[*] |
| With 0.5 band width | 8 | **0.2177** | **186**[*] |

Source: Computed based on Household Survey 2019.

**Table 8. Testing of balance of propensity score and covariates.**

| Covariates | Samples | Mean | | % bias | %reduction | t-test | P-value |
|---|---|---|---|---|---|---|---|
| | | Treated | Control | | | | |
| Gender | Unmatched | .6383 | .75532 | | | | |
| | Matched | .6383 | .58511 | 11.6 | 54.5 | 0.75 | 0.457 |
| Marital status | Unmatched | 2.1915 | 2.0745 | | | | |
| | Matched | 2.1915 | 2.1489 | 8.3 | 63.6 | 0.56 | 0.576 |
| Age | Unmatched | 45.106 | 40.989 | | | | |
| | Matched | 45.106 | 42.191 | 18.1 | 29.2 | 1.16 | 0.249 |
| Education | Unmatched | 1.3936 | 1.3936 | | | | |
| | Matched | 1.3936 | 1.4255 | -6.5 | 0.0 | -0.44 | 0.658 |
| Family size | Unmatched | 4.2021 | 4.1064 | | | | |
| | Matched | 4.2021 | 3.6383 | 30.0 | -488.9 | 2.35 | 0.020 |
| Labour | Unmatched | 2.617 | 2.8723 | | | | |
| | Matched | 2.617 | 2.6702 | -2.9 | 79.2 | -0.34 | 0.736 |
| Land size | Unmatched | .79787 | 1.1489 | | | | |
| | Matched | .79787 | 1.0106 | -25.7 | 39.4 | -1.95 | 0.053 |
| Oxen number | Unmatched | .59574 | .89362 | | | | |
| | Matched | .59574 | .57447 | 2.3 | 92.9 | 0.19 | 0.848 |
| TLU | Unmatched | 1.6277 | 2.7234 | | | | |
| | Matched | 1.6277 | 3.4787 | -69.4 | -68.9 | -4.58 | 0.000 |
| Access to credit | Unmatched | .42553 | .23404 | | | | |
| | Matched | .42553 | .41489 | 2.3 | 94.4 | 0.15 | 0.883 |
| Access to-Exten. | Unmatched | .81915 | .75532 | | | | |
| | Matched | .81915 | .58511 | 57.1 | -266.7 | 3.61 | 0.000 |
| Off-farm activity | Unmatched | .44681 | .2234 | | | | |
| | Matched | .44681 | .29787 | 32.3 | 33.3 | 2.13 | 0.035 |
| Food shortage-month | Unmatched | 4.234 | 3.3511 | | | | |
| | Matched | 4.234 | 3.1064 | 83.1 | -27.7 | 5.62 | 0.000 |

Source: Computed based on Household Survey 2019.

comparison groups. Hence, these results can be used to evaluate the impact of PSNP participation among the group of households having similar observed characteristics. This enables to compare observed outcomes for PSNP participants with those of non-participant groups sharing common support.

## Average treatment effect estimation of the impact of PSNP on households' food security

This section presents and discusses the estimation result of the impact of PSNP on households' food security using a kernel matching algorithm. The researcher used ATT and t- value

**Table 9. Chi-square test for the joint significance of variables.**

| Sample | PsR$^2$ | LR chi$^2$ | P>ch$^2$ |
|---|---|---|---|
| Unmatched | 0.215 | 56.11 | 0.000 |
| Matched | 0.265 | 69.00 | 0.000 |

Source: Computed based on Household Survey 2019.

**Table 10. Impact of PSNP participation on household consumption using ATT.**

| Outcome variable | Sample | Treated | Control | Difference | S.E | T-Stat |
|---|---|---|---|---|---|---|
| Caloric intake | ATT | 2519.29348 | 2111.64439 | 407.649088 | 159.548886 | 2.56 |

Source: Computed based on Household Survey 2019.

columns to evaluate the impact indicators. It was generally hypothesized that participating in PSNP increases household food security. Table 10 shows the ATT results for participation in productive safety net programs based on kernel matching methods. The results indicate that being a member of PSNP significantly increases household food security status. The estimated evidence showed that there is supportive evidence of a statistically significant effect on outcome variables. The result is interpreted as the impact of PSNP on household food security for PSNP program participants is 2519.29348kilocalories as compared to non-participants.

This finding is consistent with what has been studied before [9, 28, 50, 51]. Besides, many of the beneficiaries, during the focus group discussion stated that selling of land and giving the land for share (sharecropping) has almost stopped after the coming of the PSNP.

Different recent studies [8, 15, 52–55] conducted in the rural communities of Ethiopia show the positive impact of PSNP. PSNP has a positive effect on children by providing short-term nutritional benefits [52]. Recent studies indicates that PSNP has a positive impact by preventing households from selling productive assets for their consumption [55]. Specifically, PSNP has a positive effect on consumption [15, 53]. However, some scholars [19, 25, 56–60] found out that PSNP has a negative impact on the rural communities. This implies that PSNP implementation was to be multi-dimensional, which varies from place to place. For example, some communities develop a sense of dependency syndrome [57], and as a result, sell their assets, rent their land to be part of the PSNP. Therefore, in good implementation, PSNP helps beneficiaries for consumption smoothing and asset accumulation. The PSNP has a significant impact on households in *Sire* District.

## Checking robustness of average treatment effect on food security

The strength of the propensity score matching model for the average treatment effect on treated households was found to be good. Based on Rosenbaun, Nnmatch was used to check the robustness of the ATT for the outcome variables which is household food security [45]. As seen in Table 11, independent variables used to estimate the outcome variable food security

**Table 11. Average treatment effect for the treated-on food security.**

| Matching estimator: Average Treatment Effect for the Treated | | | | | |
|---|---|---|---|---|---|
| Weighting matrix: inverse variance Number of obs = 188 | | | | | |
| | | | | | Number of matches (m) = 1 |
| Food security | Coef. | Std. Err. | Z | P>\|z\| | [95% Conf. Interval] |
| SATT | 497.117 | 229.333 | 2.17 | 0.030 | 47.6326 946.6014 |

Source: Computed based on Household Survey 2019.

are found to be, jointly, statistically significant with Z value 2.17 and P value less than one percent. The result aligns with the ATT result of food security we did.

## Sensitivity analysis

The PSM approach cannot be fully controlled for unobservable bias. As is suggested by Rosenbaum and Rubin, Gebrehiwot and Van der, the presentation of matching estimates should go with Sensitivity analysis [45, 61]. Thus, the study verified the sensitivity of the estimated treatment effects on selection on the unobservable bias using the bounding approach designed by Rosenbaum and Rubin [45].

This procedure used the matching estimates to determine the confidence interval of the outcome variable for different values of μ(gamma). Gamma captures the level of association of unobserved characteristics with the treatment and outcome required for it(the unobserved characteristics) to explain the observed impacts. If the lowest μ, which encompasses 0, then one may state that the probability of such unobserved bias is relatively high and the estimated impact is therefore sensitive to the existence of unobservable [45]. Table 12 reports the upper and lower bounds results, showing that under the assumption of no hidden bias, when μ = 1, the +sig- test statistics indicates a highly significant treatment effect for PSNP intervention on household food calorie intake. The effect is significant under μ = 1 if we have underestimated the true treatment effect. The Sig+ revealed that the study is insensitive to hidden bias. The sensitivity analysis, therefore, indicates that the observed results on the impact of PSNP on households' food calorie intakes were robust and the results were not sensitive to confounders (Table 12).

## Conclusions and policy implications

The main idea of this study was to answer the question, "What could happen if the program was not in place "in *Sire* District. To answer this and come up with the final result, the Propensity score matching method was applied. The findings of this research based on the estimation result of the Average Treatment Effect on the Treated (ATT) indicates that implementation of PSNP had an impact on households food security status. The result of sensitivity analysis shows that estimated ATT for calorie intake (the outcome variable) is insensitive which indicates its robustness.

The beneficiaries' households were more food secure than non-beneficiary households by 68% and54% respectively. The mean energy available for the beneficiary and non-beneficiary households is 2488.500 and 2153.394 kcal, respectively. Estimates of the average treatment

**Table 12. Food security sensitivity analysis table.**

| Gama | Sig+ | Sig- | t-hat+ | t-hat- | CI+ | CI- |
|------|------|------|--------|--------|-----|-----|
| 1 | .00209 | .002409 | 408 | 408 | 116.5 | 700 |
| 1.05 | .004469 | .00001242 | 379.5 | 438.5 | 92.001 | 734.5 |
| 1.1 | .007743 | .0000633 | 350 | 461 | 66 | 763 |
| 1.15 | .012642 | .0000319 | 315.5 | 488.5 | 40 | 795.5 |
| 1.2 | .019598 | .000016 | 286.5 | 519.5 | 13.001 | 820 |
| 1.25 | .029031 | .000079 | 260.5 | 546.5 | -12.5 | 850 |
| 1.3 | .041314 | .000039 | 238.5 | 563 | -38.5001 | 869.5 |
| 1.35 | .056745 | .000019 | 213.5 | 595.25 | -58.5 | 897 |

Source: Computed based on Household Survey 2019.

effect of the treated indicated that the Productive Safety Net Program can impact households' food security by 2519.2938 kilocalories higher in the kernel matching method. Based on study findings, future policy recommendations were developed: The study found that PSNP had significantly increased households calorie intake. This is an encouraging indicator for Programme implementers and funding agents.

- Therefore, channeling further efforts on this indicator is important for a more pronounced impact of the Programme.

- The government, non-governmental organizations and other concerned bodies should emphasize strengthening the program execution and insisting local governments involve vulnerable households like women in the PSNP to ensure their food security.

- Projects aim at nutrition and consumption should be encouraged by the government in the rural communities including study areas.

- In addition, the government and non-government organizations should strengthen agricultural education as a development intervention. This would help rural households to have better food consumption habits and make them in better food security status. Hence, there are variations of program impact studies in different countries, this study recommends further research with greater scope and at different locations on the impact of PSNP and other Food Security programs on household food security.

## Supporting information

**S1 File.**
(ZIP)

## Acknowledgments

The authors would like to acknowledge the rural communities of *Sire* District for their time during data collection and those individuals who gave us comments in the first draft of the manuscript.

## Author Contributions

**Data curation:** Aregash Getachew Hailu, Zerihun Yohannes Amare.

**Formal analysis:** Aregash Getachew Hailu, Zerihun Yohannes Amare.

**Funding acquisition:** Aregash Getachew Hailu.

**Methodology:** Aregash Getachew Hailu, Zerihun Yohannes Amare.

**Software:** Aregash Getachew Hailu.

**Supervision:** Aregash Getachew Hailu.

**Validation:** Aregash Getachew Hailu, Zerihun Yohannes Amare.

**Visualization:** Zerihun Yohannes Amare.

**Writing – original draft:** Aregash Getachew Hailu.

**Writing – review & editing:** Zerihun Yohannes Amare.

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
