## [Decision Letter · Decision Letter 0]

22 Apr 2021

PONE-D-21-07822

Impact of productive safety net program on food security of beneficiary households in western Ethiopia: A matching estimator approach

PLOS ONE

Dear Dr. Aregash Getachew Hailu,

Thank you for submitting your manuscript to PLOS ONE. After careful consideration, we feel that it has merit but does not fully meet PLOS ONE’s publication criteria as it currently stands. Therefore, we invite you to submit a revised version of the manuscript that addresses the points raised during the review process.

We look forward to receiving your revised manuscript.

Kind regards,

László VASA, PhD

Academic Editor

PLOS ONE

Journal Requirements:

PLOS requires an ORCID iD for the corresponding author in Editorial Manager on papers submitted after December 6th, 2016. Please ensure that you have an ORCID iD and that it is validated in Editorial Manager. To do this, go to ‘Update my Information’ (in the upper left-hand corner of the main menu), and click on the Fetch/Validate link next to the ORCID field. This will take you to the ORCID site and allow you to create a new iD or authenticate a pre-existing iD in Editorial Manager. Please see the following video for instructions on linking an ORCID iD to your Editorial Manager account: https://www.youtube.com/watch?v=_xcclfuvtxQ

We suggest you thoroughly copyedit your manuscript for language usage, spelling, and grammar. If you do not know anyone who can help you do this, you may wish to consider employing a professional scientific editing service. 

We note you have included a table to which you do not refer in the text of your manuscript. Please ensure that you refer to Table 3, 10,  12, in your text; if accepted, production will need this reference to link the reader to the Table.

Please ensure that you refer to Figure 1 in your text as, if accepted, production will need this reference to link the reader to the figure.

7. Thank you for submitting the above manuscript to PLOS ONE. During our internal evaluation of the manuscript, we found significant text overlap between your submission and the following previously published works:

- https://www.duo.uio.no/bitstream/handle/10852/44340/final-version.pdf?isAllowed=y&sequence=1

- https://nadre.ethernet.edu.et/record/983?ln=en#.YIIOFpBKg2w

Please revise the manuscript to rephrase the duplicated text, cite your sources, and provide details as to how the current manuscript advances on previous work. Please note that further consideration is dependent on the submission of a manuscript that addresses these concerns about the overlap in text with published work.

Reviewers' comments:

Reviewer's Responses to Questions

**Comments to the Author**

1. Is the manuscript technically sound, and do the data support the conclusions?

Reviewer #1: Yes

Reviewer #2: Yes

Reviewer #3: Yes

2. Has the statistical analysis been performed appropriately and rigorously? 

Reviewer #1: Yes

Reviewer #2: Yes

Reviewer #3: Yes

3. Have the authors made all data underlying the findings in their manuscript fully available?

Reviewer #1: Yes

Reviewer #2: Yes

Reviewer #3: Yes

4. Is the manuscript presented in an intelligible fashion and written in standard English?

Reviewer #1: Yes

Reviewer #2: No

Reviewer #3: No

5. Review Comments to the Author

Reviewer #1: The topic of is interesting and relevant. The elaboration and the applied methodology is accurate, however a formal deficiency has to be highlighted: at the tables and figures there are no source indication which is a necessary requirement in case of scientific papers. It can be seen that most probably all (or the majority) of data derived from the primary research of the authors, but even if so, it has to be indicated accordingly.

The bibliographic review could be a bit broadened especially a part of the timely issues and challenges of food supply security in Ethiopia and possibly in some other sub-Saharan African countries to make the context more international. The international bibliography is rich in this respect (e.g. https://pubs.iied.org/sites/default/files/pdfs/migrate/14640IIED.pdf or https://re.volsu.ru/eng/contacts/8_Nad_i_dr.pmd.pdf .

The authors are also advised to specify and underline the applicability of their new results and findings possibly in a structured list of recommendations. After the suggested improvements were made, the paper can be recommended to be accepted and published.

Reviewer #2: The manuscript is technically sound, the data the authors have collected supports the study and the conclusions, thus the study presents the results of original research. What was especially welcomed is the critical approach towards similar program evaluations in Ethiopia and the fact that these previous studies did not include non-beneficiary households in their researches, thus relying on the beneficiary households to interpret their former situation, many times ending up with bias reporting, as the authors put it "respondents recall is often incorrect since it is hard to remember all past incidents correctly, resulting in over or under-reporting of past incidents". According to the submitted information, the authors have not published this study elsewhere. The structure of the paper is logical, starts with introducing the situation first and then presenting the methods and sources of data collection. In this case 188 households were included in the survey, cross-sectional survey data were gathered from these, further key informants interviews and focus group discussion was employed to triangulate results and a Chi-square test was employed to compare the households’ food security status. From the paper we can see that interviews and other analyses are performed to a high technical standard and are described in sufficient detail within the paper, the statistical analysis been performed appropriately and rigorously. The authors have made all data underlying the findings in their manuscript fully available within tables, explaining those tables as well. Also the conclusion is presented in the required way and data supports the findings, the policy implications are logical and supported by findings. The research meets all applicable standards for the ethics of experimentation and research integrity and the paper adheres to appropriate reporting guidelines and community standards for data availability. The only issue with the recent form of the paper is that it is not presented in an intelligible fashion and a language check is required as well. There are many typos within the text, missing spaces and the incorrect use of quotation marks (not where they are used but are cases where not the quotation mark is used for a citation or the citation is not closed correctly) make it harder to enjoy the paper. My recommendation is to publish the paper after the minor revisions regarding the correction of the language and style mistakes.

Reviewer #3: The topic itself is really actual, even nowadays, despite the efforts of the country to deal with food secutity problems. The paper as a whole, the theme would be fine for publishing in PLOS ONE, however, in details, some modifications are needed.

- Introduction should be more focused and better structured (highlighting the research questions more clearly, context, objective)

- In fact, there is no literature review while it is a must. Authors should write a comprehensive, analytical and critical literature review chapter. In the paper the essential international literature dealing with this topic and/or focusing on Ethiopia should be reviewed. Other countres' cases could be interesting as well (e.g. https://doi.org/10.21163%2FGT_2020.152.16 or the issue of trust is described well https://doi.org/10.1080/08974438.2013.833567)

- Metodology chapter should be better focused and more clear, now it seems like unstructured.

- English proofreading is needed to provide the expected scientific and right English.

6. PLOS authors have the option to publish the peer review history of their article (what does this mean?). If published, this will include your full peer review and any attached files.

Reviewer #1: No

Reviewer #2: No

Reviewer #3: No

---

## [Author Response · Author response to Decision Letter 0]

16 Jun 2021

Attached in a separate paper. You can also see here

Response: The authors tried to meet the PLOS ONE’s style requirements. 

Response: The authors has been submitted to professional editors and got revised manuscript

. Address of the editor: 

ABOLARIN Gbminiyi, Ph.D. 

abolaring@babcock.edu.ng

Editor, College of Postgraduate Studies, 

Babcock University, Ogun State, Nigeria

3. PLOS requires an ORCID iD for the corresponding author in Editorial Manager 

Response: 

ORCID-Aregash Getachew Hailu

https://orcid.org/0000-0002-4822-0470

ORCID-Zerihun Yohannes Amare 

https://orcid.org/0000-0002-4503-6790

● The name of the colleague or the details of the professional service that edited your manuscript: see the full details of the professional editor 

● A copy of your manuscript showing your changes by either highlighting them or using track changes (uploaded as a *supporting information* file)---Attached 

Response: Attached 

1. We note you have included a table to which you do not refer in the text of your manuscript. Please ensure that you refer to Table 3, 10, 12, in your text; if accepted, production will need this reference to link the reader to the Table.

Response: Corrected 

2. Please ensure that you refer to Figure 1 in your text as, if accepted, production will need this reference to link the reader to the figure.

Response: Corrected 

Response: corrected 

7. Thank you for submitting the above manuscript to PLOS ONE. During our internal evaluation of the manuscript, we found significant text overlap between your submission and the following previously published works:

Please revise the manuscript to rephrase the duplicated text, cite your sources, and provide details as to how the current manuscript advances on previous work. Please note that further consideration is dependent on the submission of a manuscript that addresses these concerns about the overlap in text with published work.

Response: 

Reviewers' comments:

Reviewer's Responses to Questions

Comments to the Author

1. Is the manuscript technically sound, and do the data support the conclusions?

Reviewer #1: Yes

Reviewer #2: Yes

Reviewer #3: Yes

2. Has the statistical analysis been performed appropriately and rigorously?

Reviewer #1: Yes

Reviewer #2: Yes

Reviewer #3: Yes

3. Have the authors made all data underlying the findings in their manuscript fully available?

Reviewer #1: Yes

Reviewer #2: Yes

Reviewer #3: Yes

4. Is the manuscript presented in an intelligible fashion and written in standard English?

Reviewer #1: Yes

Reviewer #2: No

Reviewer #3: No

Response: The language is edited as to the level of standard with accredited editor. The address is described at the beginning of this document. 

5. Review Comments to the Author

Reviewer #1: The topic of is interesting and relevant. The elaboration and the applied methodology is accurate, however a formal deficiency has to be highlighted: at the tables and figures there are no source indication which is a necessary requirement in case of scientific papers. It can be seen that most probably all (or the majority) of data derived from the primary research of the authors, but even if so, it has to be indicated accordingly.

Response: proofread and Corrected:

The bibliographic review could be a bit broadened especially a part of the timely issues and challenges of food supply security in Ethiopia and possibly in some other sub-Saharan African countries to make the context more international. The international bibliography is rich in this respect (e.g. https://pubs.iied.org/sites/default/files/pdfs/migrate/14640IIED.pdf or https://re.volsu.ru/eng/contacts/8_Nad_i_dr.pmd.pdf .

The authors are also advised to specify and underline the applicability of their new results and findings possibly in a structured list of recommendations. After the suggested improvements were made, the paper can be recommended to be accepted and published.

Response: the suggested and other literatures are included 

Reviewer #2: The manuscript is technically sound, the data the authors have collected supports the study and the conclusions, thus the study presents the results of original research. What was especially welcomed is the critical approach towards similar program evaluations in Ethiopia and the fact that these previous studies did not include non-beneficiary households in their researches, thus relying on the beneficiary households to interpret their former situation, many times ending up with bias reporting, as the authors put it "respondents recall is often incorrect since it is hard to remember all past incidents correctly, resulting in over or under-reporting of past incidents". According to the submitted information, the authors have not published this study elsewhere. The structure of the paper is logical, starts with introducing the situation first and then presenting the methods and sources of data collection. In this case 188 households were included in the survey, cross-sectional survey data were gathered from these, further key informants interviews and focus group discussion was employed to triangulate results and a Chi-square test was employed to compare the households’ food security status. From the paper we can see that interviews and other analyses are performed to a high technical standard and are described in sufficient detail within the paper, the statistical analysis been performed appropriately and rigorously. The authors have made all data underlying the findings in their manuscript fully available within tables, explaining those tables as well. Also the conclusion is presented in the required way and data supports the findings, the policy implications are logical and supported by findings. The research meets all applicable standards for the ethics of experimentation and research integrity and the paper adheres to appropriate reporting guidelines and community standards for data availability. The only issue with the recent form of the paper is that it is not presented in an intelligible fashion and a language check is required as well. There are many typos within the text, missing spaces and the incorrect use of quotation marks (not where they are used but are cases where not the quotation mark is used for a citation or the citation is not closed correctly) make it harder to enjoy the paper. My recommendation is to publish the paper after the minor revisions regarding the correction of the language and style mistakes.

Response: The authors tried to improve the presentation styles and the language was also proofread by accredited editor. 

Reviewer #3: The topic itself is really actual, even nowadays, despite the efforts of the country to deal with food secutity problems. The paper as a whole, the theme would be fine for publishing in PLOS ONE, however, in details, some modifications are needed.

- Introduction should be more focused and better structured (highlighting the research questions more clearly, context, objective)

- In fact, there is no literature review while it is a must. Authors should write a comprehensive, analytical and critical literature review chapter. In the paper the essential international literature dealing with this topic and/or focusing on Ethiopia should be reviewed. Other countres' cases could be interesting as well (e.g. https://doi.org/10.21163%2FGT_2020.152.16 or the issue of trust is described well https://doi.org/10.1080/08974438.2013.833567)

- Metodology chapter should be better focused and more clear, now it seems like unstructured.

- English proofreading is needed to provide the expected scientific and right English.

 Response: The authors tried to maintain the journal formalities and include the literature as suggested.

---

## [Decision Letter · Decision Letter 1]

11 Jul 2021

PONE-D-21-07822R1

Impact of productive safety net program on food security of beneficiary households in western Ethiopia: A matching estimator approach

PLOS ONE

Dear Dr. Aregash Getachew Hailu,

Thank you for submitting your manuscript to PLOS ONE. After careful consideration, we feel that it has merit but does not fully meet PLOS ONE’s publication criteria as it currently stands. Therefore, we invite you to submit a revised version of the manuscript that addresses the points raised during the review process.

We look forward to receiving your revised manuscript.

Kind regards,

László Vasa, PhD

Academic Editor

PLOS ONE

Journal Requirements:

Reviewers' comments:

Reviewer's Responses to Questions

**Comments to the Author**

1. If the authors have adequately addressed your comments raised in a previous round of review and you feel that this manuscript is now acceptable for publication, you may indicate that here to bypass the “Comments to the Author” section, enter your conflict of interest statement in the “Confidential to Editor” section, and submit your "Accept" recommendation.

Reviewer #1: (No Response)

Reviewer #3: (No Response)

2. Is the manuscript technically sound, and do the data support the conclusions?

Reviewer #1: Yes

Reviewer #3: Partly

3. Has the statistical analysis been performed appropriately and rigorously? 

Reviewer #1: Yes

Reviewer #3: Yes

4. Have the authors made all data underlying the findings in their manuscript fully available?

Reviewer #1: Yes

Reviewer #3: Yes

5. Is the manuscript presented in an intelligible fashion and written in standard English?

Reviewer #1: Yes

Reviewer #3: Yes

6. Review Comments to the Author

Reviewer #1: The bibliographic extension was partially done, however it could be still further broadened, like https://re.volsu.ru/eng/contacts/8_Nad_i_dr.pmd.pdf , https://ojs.lib.unideb.hu/apstract/article/view/6217/5834 .

The authors were also advised to specify and underline the applicability of their new results and findings in a structured list of recommendations, but it is still missing from the revised version. In my opinion, these improvements may merely increase the value of the paper. After the suggested recommendations were implemented the paper can be published.

Reviewer #3: The authors did not made the reqommended improvements so I can't accept this paper for publication in its present form. I advise to check my comments and recommendations of the first review round again.

7. PLOS authors have the option to publish the peer review history of their article (what does this mean?). If published, this will include your full peer review and any attached files.

Reviewer #1: No

Reviewer #3: No

---

## [Author Response · Author response to Decision Letter 1]

10 Aug 2021

Response: The authors tried to meet the PLOS ONE’s style requirements. 

Response: The authors has been submitted to professional editors and got revised manuscript

. Address of the editor: 

ABOLARIN Gbminiyi, Ph.D. 

abolaring@babcock.edu.ng

Editor, College of Postgraduate Studies, 

Babcock University, Ogun State, Nigeria

3. PLOS requires an ORCID iD for the corresponding author in Editorial Manager 

Response: 

ORCID-Aregash Getachew Hailu

https://orcid.org/0000-0002-4822-0470

ORCID-Zerihun Yohannes Amare 

https://orcid.org/0000-0002-4503-6790

● The name of the colleague or the details of the professional service that edited your manuscript: see the full details of the professional editor 

● A copy of your manuscript showing your changes by either highlighting them or using track changes (uploaded as a *supporting information* file)---Attached 

Response: Attached 

1. We note you have included a table to which you do not refer in the text of your manuscript. Please ensure that you refer to Table 3, 10, 12, in your text; if accepted, production will need this reference to link the reader to the Table.

Response: Corrected 

2. Please ensure that you refer to Figure 1 in your text as, if accepted, production will need this reference to link the reader to the figure.

Response: Corrected 

Response: corrected 

7. Thank you for submitting the above manuscript to PLOS ONE. During our internal evaluation of the manuscript, we found significant text overlap between your submission and the following previously published works:

Please revise the manuscript to rephrase the duplicated text, cite your sources, and provide details as to how the current manuscript advances on previous work. Please note that further consideration is dependent on the submission of a manuscript that addresses these concerns about the overlap in text with published work.

Response: 

Reviewers' comments:

Reviewer's Responses to Questions

Comments to the Author

1. Is the manuscript technically sound, and do the data support the conclusions?

Reviewer #1: Yes

Reviewer #2: Yes

Reviewer #3: Yes

2. Has the statistical analysis been performed appropriately and rigorously?

Reviewer #1: Yes

Reviewer #2: Yes

Reviewer #3: Yes

3. Have the authors made all data underlying the findings in their manuscript fully available?

Reviewer #1: Yes

Reviewer #2: Yes

Reviewer #3: Yes

4. Is the manuscript presented in an intelligible fashion and written in standard English?

Reviewer #1: Yes

Reviewer #2: No

Reviewer #3: No

Response: The language is edited as to the level of standard with accredited editor. The address is described at the beginning of this document. 

5. Review Comments to the Author

Reviewer #1: The topic of is interesting and relevant. The elaboration and the applied methodology is accurate, however a formal deficiency has to be highlighted: at the tables and figures there are no source indication which is a necessary requirement in case of scientific papers. It can be seen that most probably all (or the majority) of data derived from the primary research of the authors, but even if so, it has to be indicated accordingly.

Response: proofread and Corrected:

The bibliographic review could be a bit broadened especially a part of the timely issues and challenges of food supply security in Ethiopia and possibly in some other sub-Saharan African countries to make the context more international. The international bibliography is rich in this respect (e.g. https://pubs.iied.org/sites/default/files/pdfs/migrate/14640IIED.pdf or https://re.volsu.ru/eng/contacts/8_Nad_i_dr.pmd.pdf .

The authors are also advised to specify and underline the applicability of their new results and findings possibly in a structured list of recommendations. After the suggested improvements were made, the paper can be recommended to be accepted and published.

Response: the suggested and other literatures are included 

Reviewer #2: The manuscript is technically sound, the data the authors have collected supports the study and the conclusions, thus the study presents the results of original research. What was especially welcomed is the critical approach towards similar program evaluations in Ethiopia and the fact that these previous studies did not include non-beneficiary households in their researches, thus relying on the beneficiary households to interpret their former situation, many times ending up with bias reporting, as the authors put it "respondents recall is often incorrect since it is hard to remember all past incidents correctly, resulting in over or under-reporting of past incidents". According to the submitted information, the authors have not published this study elsewhere. The structure of the paper is logical, starts with introducing the situation first and then presenting the methods and sources of data collection. In this case 188 households were included in the survey, cross-sectional survey data were gathered from these, further key informants interviews and focus group discussion was employed to triangulate results and a Chi-square test was employed to compare the households’ food security status. From the paper we can see that interviews and other analyses are performed to a high technical standard and are described in sufficient detail within the paper, the statistical analysis been performed appropriately and rigorously. The authors have made all data underlying the findings in their manuscript fully available within tables, explaining those tables as well. Also the conclusion is presented in the required way and data supports the findings, the policy implications are logical and supported by findings. The research meets all applicable standards for the ethics of experimentation and research integrity and the paper adheres to appropriate reporting guidelines and community standards for data availability. The only issue with the recent form of the paper is that it is not presented in an intelligible fashion and a language check is required as well. There are many typos within the text, missing spaces and the incorrect use of quotation marks (not where they are used but are cases where not the quotation mark is used for a citation or the citation is not closed correctly) make it harder to enjoy the paper. My recommendation is to publish the paper after the minor revisions regarding the correction of the language and style mistakes.

Response: The authors tried to improve the presentation styles and the language was also proofread by accredited editor. 

Reviewer #3: The topic itself is really actual, even nowadays, despite the efforts of the country to deal with food secutity problems. The paper as a whole, the theme would be fine for publishing in PLOS ONE, however, in details, some modifications are needed.

- Introduction should be more focused and better structured (highlighting the research questions more clearly, context, objective)

- In fact, there is no literature review while it is a must. Authors should write a comprehensive, analytical and critical literature review chapter. In the paper the essential international literature dealing with this topic and/or focusing on Ethiopia should be reviewed. Other countres' cases could be interesting as well (e.g. https://doi.org/10.21163%2FGT_2020.152.16 or the issue of trust is described well https://doi.org/10.1080/08974438.2013.833567)

- Metodology chapter should be better focused and more clear, now it seems like unstructured.

- English proofreading is needed to provide the expected scientific and right English.

 Response: The authors tried to maintain the journal formalities and include the literature as suggested. The research questions and objectives are listed in a structured way. 

Second time reviewers’ comments 

Reviewer #1: The bibliographic extension was partially done, however it could be still further broadened, like https://re.volsu.ru/eng/contacts/8_Nad_i_dr.pmd.pdf , https://ojs.lib.unideb.hu/apstract/article/view/6217/5834 .

The authors were also advised to specify and underline the applicability of their new results and findings in a structured list of recommendations, but it is still missing from the revised version. In my opinion, these improvements may merely increase the value of the paper. After the suggested recommendations were implemented the paper can be published.

 Response: The authors appreciate the comments of the reviewer and tried to expand using recent published papers and reviewers suggested articles. 

Reviewer #3: The authors did not made the recommended improvements so I can't accept this paper for publication in its present form. I advise to check my comments and recommendations of the first review round again.

Response: We revised based on the previous comments and suggestions. If we go beyond this, and add more literature, the page number will be increased and out of the PLOS ONE submission guideline.

---

## [Decision Letter · Decision Letter 2]

31 Aug 2021

PONE-D-21-07822R2

Impact of productive safety net program on food security of beneficiary households in western Ethiopia: A matching estimator approach

PLOS ONE

Dear Dr. Hailu,

Thank you for submitting your manuscript to PLOS ONE. After careful consideration, we feel that it has merit but does not fully meet PLOS ONE’s publication criteria as it currently stands. Therefore, we invite you to submit a revised version of the manuscript that addresses the points raised during the review process.

We look forward to receiving your revised manuscript.

Kind regards,

László Vasa, PhD

Academic Editor

PLOS ONE

Journal Requirements:

Additional Editor Comments (if provided):

Reviewers' comments:

Reviewer's Responses to Questions

**Comments to the Author**

1. If the authors have adequately addressed your comments raised in a previous round of review and you feel that this manuscript is now acceptable for publication, you may indicate that here to bypass the “Comments to the Author” section, enter your conflict of interest statement in the “Confidential to Editor” section, and submit your "Accept" recommendation.

Reviewer #1: All comments have been addressed

Reviewer #2: (No Response)

Reviewer #3: All comments have been addressed

2. Is the manuscript technically sound, and do the data support the conclusions?

Reviewer #1: Yes

Reviewer #2: Yes

Reviewer #3: Yes

3. Has the statistical analysis been performed appropriately and rigorously? 

Reviewer #1: Yes

Reviewer #2: Yes

Reviewer #3: Yes

4. Have the authors made all data underlying the findings in their manuscript fully available?

Reviewer #1: Yes

Reviewer #2: Yes

Reviewer #3: Yes

5. Is the manuscript presented in an intelligible fashion and written in standard English?

Reviewer #1: No

Reviewer #2: No

Reviewer #3: Yes

6. Review Comments to the Author

Reviewer #1: The Authors repeatedly disregarded the reviewer's advise. The smaller problem is that they did not incorporate the suggested sources into the bibliographic review, but the structured and possibly detailed chapter of conclusions and suggestions is still missing (even though the original chapter was slightly extended). This part could have highlighted also the new scientific findings of the paper. Another - formal - problem is that the authors did not indicate the sources of data in case of most Tables in the paper. It is very much necessary in case of scientific papers, even if the data were eventually collected by the authors' primary research.

Reviewer #2: After reading the new submission (corrected version) i still can see many of those mistakes which are not acceptable for an academic journal. There are still many typos within the text, the ones which have been highlighted already (missing spaces and the incorrect use of quotation marks). These has to be corrected before acceptance.

Reviewer #3: The authors accapted all of my recommendations for improving the paper and modified/upgraded it in the right way. The paper is more sound in its current form, so I recommend it for publications without changes.

7. PLOS authors have the option to publish the peer review history of their article (what does this mean?). If published, this will include your full peer review and any attached files.

Reviewer #1: No

Reviewer #2: No

Reviewer #3: No

---

## [Author Response · Author response to Decision Letter 2]

11 Oct 2021

Reviewer #1: The Authors repeatedly disregarded the reviewer's advise. The smaller problem is that they did not incorporate the suggested sources into the bibliographic review, but the structured and possibly detailed chapter of conclusions and suggestions is still missing (even though the original chapter was slightly extended). This part could have highlighted also the new scientific findings of the paper. Another - formal - problem is that the authors did not indicate the sources of data in case of most Tables in the paper. It is very much necessary in case of scientific papers, even if the data were eventually collected by the authors' primary research.

Authors’ response: 1.The “reviewer one” suggested to include these two unrelated papers:

 1.1 https://re.volsu.ru/eng/contacts/8_Nad_i_dr.pmd.pdf

 1.2. https://ojs.lib.unideb.hu/apstract/article/view/6217/5834

These suggested materials are not related to our article and we are unable to add. 

2. Conclusion and recommendations are already corrected based on the first session reviewers comments and you can check the first submission of the manuscript with track change 

3. The sources of the data are cited in all Tables and acknowledged. 

Reviewer #2: After reading the new submission (corrected version) i still can see many of those mistakes which are not acceptable for an academic journal. There are still many typos within the text, the ones which have been highlighted already (missing spaces and the incorrect use of quotation marks). These has to be corrected before acceptance.

Response: all typos and inappropriate quotation marks are corrected 

Reviewer #3: The authors accapted all of my recommendations for improving the paper and modified/upgraded it in the right way. The paper is more sound in its current form, so I recommend it for publications without changes.

---

## [Decision Letter · Decision Letter 3]

3 Nov 2021

PONE-D-21-07822R3Impact of productive safety net program on food security of beneficiary households in western Ethiopia: A matching estimator approachPLOS ONE

Dear Dr. Hailu,

Thank you for submitting your manuscript to PLOS ONE. After careful consideration, we feel that it has merit but does not fully meet PLOS ONE’s publication criteria as it currently stands. Therefore, we invite you to submit a revised version of the manuscript that addresses the points raised during the review process.

We look forward to receiving your revised manuscript.

Kind regards,

László Vasa, PhD

Academic Editor

PLOS ONE

Journal Requirements:

Reviewers' comments:

Reviewer's Responses to Questions

**Comments to the Author**

1. If the authors have adequately addressed your comments raised in a previous round of review and you feel that this manuscript is now acceptable for publication, you may indicate that here to bypass the “Comments to the Author” section, enter your conflict of interest statement in the “Confidential to Editor” section, and submit your "Accept" recommendation.

Reviewer #2: (No Response)

Reviewer #3: All comments have been addressed

2. Is the manuscript technically sound, and do the data support the conclusions?

Reviewer #2: Yes

Reviewer #3: Yes

3. Has the statistical analysis been performed appropriately and rigorously? 

Reviewer #2: Yes

Reviewer #3: Yes

4. Have the authors made all data underlying the findings in their manuscript fully available?

Reviewer #2: Yes

Reviewer #3: Yes

5. Is the manuscript presented in an intelligible fashion and written in standard English?

Reviewer #2: No

Reviewer #3: Yes

6. Review Comments to the Author

Reviewer #2: After reading the submission (the newly submitted version), though the authors claimed they have crrected everything, i still can see many mistakes which are, in my understanding, not acceptable for an academic journal. There are still many typos within the text, the ones which have been highlighted already (missing spaces and the incorrect use of quotation marks). These has to be corrected before acceptance.

Just a few examples to help the authors:

Table 9 is numbered as Table 1.

No coherency in naming the tables, See: 'Table 11:' vs. 'Table 12' (few times with a colon, sometimes without it)

Introduction part for example with missing spaces: 'Africa(Gollin 2014)'; 'households(FAO 2017)'; '(2011)impact'; 'world(United Nations 2000)'

These are all over the text.

Reviewer #3: The authors improved the paper in accordance with the reviewers' recommendations. After several rounds, the paper is now eligible for the PLOS ONE standard and can be published without any further changes.

7. PLOS authors have the option to publish the peer review history of their article (what does this mean?). If published, this will include your full peer review and any attached files.

Reviewer #2: No

Reviewer #3: No

---

## [Author Response · Author response to Decision Letter 3]

10 Nov 2021

Response to reviewer 

Reviewer #2: After reading the submission (the newly submitted version), though the authors claimed they have corrected everything, i still can see many mistakes which are, in my understanding, not acceptable for an academic journal. There are still many typos within the text, the ones which have been highlighted already (missing spaces and the incorrect use of quotation marks). These has to be corrected before acceptance.

Response: We tried to read seriously and address all the comments. See the track change for evidence

---

## [Editor Report · Decision Letter 4]

18 Nov 2021

Impact of productive safety net program on food security of beneficiary households in western Ethiopia: A matching estimator approach

PONE-D-21-07822R4

Dear Dr. Hailu,

We’re pleased to inform you that your manuscript has been judged scientifically suitable for publication and will be formally accepted for publication once it meets all outstanding technical requirements.

Kind regards,

László Vasa, PhD

Academic Editor

PLOS ONE
---

## [Editor Report · Acceptance letter]

14 Dec 2021

PONE-D-21-07822R4 

Impact of productive safety net program on food security of beneficiary households in western Ethiopia: A matching estimator approach 

Dear Dr. Hailu:

I'm pleased to inform you that your manuscript has been deemed suitable for publication in PLOS ONE. Congratulations! Your manuscript is now with our production department. 

Kind regards, 

on behalf of

Prof. Dr. László Vasa 

Academic Editor

PLOS ONE